# Single-molecule view of coordination in a multi-functional DNA polymerase

**Raymond F Pauszek III, Rajan Lamichhane[†], Arishma Rajkarnikar Singh, David P Millar***

Department of Integrative Structural and Computational Biology, The Scripps Research Institute, La Jolla, United States

**Abstract** Replication and repair of genomic DNA requires the actions of multiple enzymatic functions that must be coordinated in order to ensure efficient and accurate product formation. Here, we have used single-molecule FRET microscopy to investigate the physical basis of functional coordination in DNA polymerase I (Pol I) from *Escherichia coli*, a key enzyme involved in lagging-strand replication and base excision repair. Pol I contains active sites for template-directed DNA polymerization and 5' flap processing in separate domains. We show that a DNA substrate can spontaneously transfer between polymerase and 5' nuclease domains during a single encounter with Pol I. Additionally, we show that the flexibly tethered 5' nuclease domain adopts different positions within Pol I-DNA complexes, depending on the nature of the DNA substrate. Our results reveal the structural dynamics that underlie functional coordination in Pol I and are likely relevant to other multi-functional DNA polymerases.

**\*For correspondence:**
millar@scripps.edu

**Present address:** [†]Department of Biochemistry and Cellular and Molecular Biology, The University of Tennessee, Knoxville, United States

**Competing interests:** The authors declare that no competing interests exist.

## Introduction

DNA polymerases from many organisms need to coordinate multiple enzymatic activities to achieve accurate and efficient replication and repair of DNA, while avoiding the formation of mutagenic or unstable DNA intermediates (*Reha-Krantz, 2010*; *Bębenek and Ziuzia-Graczyk, 2018*). DNA polymerase I (Pol I), a key enzyme involved in DNA replication and repair in *Escherichia coli* (*Makiela-Dzbenska et al., 2009*; *Imai et al., 2007*; *Patel et al., 2001*), contains three distinct enzymatic activities in a single 928 residue polypeptide: a DNA-dependent 5'–3' polymerase (*pol*), a proofreading 3'–5' exonuclease (*exo*) and a 5' nuclease (*5' nuc*) (*Kelley and Joyce, 1983*; *Setlow and Kornberg, 1972*). The *pol* and *exo* activities are contained in separate domains, which together comprise the main body of the enzyme, whereas the *5' nuc* activity is located in an independent domain that is tethered to the body by an unstructured 16 amino acid (aa) peptide linker. The *5' nuc* domain is related to the flap endonuclease (FEN) family of structure-specific DNA nucleases (*Harrington and Lieber, 1994*).

Pol I plays an important role in lagging strand DNA replication in *E. coli* (*Balakrishnan and Bambara, 2013*; *Okazaki et al., 1971*). During this complex process, short RNA primers anneal to the lagging strand and are extended by DNA primase, producing fused RNA-DNA strands (Okazaki fragments). The nascent DNA portion is subsequently extended by Pol I, until the growing strand encounters another Okazaki fragment lying downstream, displacing the 5' end and forming an RNA flap. The *5' nuc* activity of Pol I then cleaves the RNA flap, generating a nicked DNA substrate that is subsequently sealed by a DNA ligase (*Figure 1A*). The same processing steps are also performed by Pol I during DNA base excision repair, in which case the displaced strand is composed of DNA (*Imai et al., 2007*).

During either Okazaki fragment processing or base excision repair, Pol I must achieve an appropriate balance between the *pol* and *5' nuc* activities in order to generate a nicked duplex product that can subsequently be sealed by a DNA ligase (*Mortusewicz et al., 2006*). Excessive *pol* activity

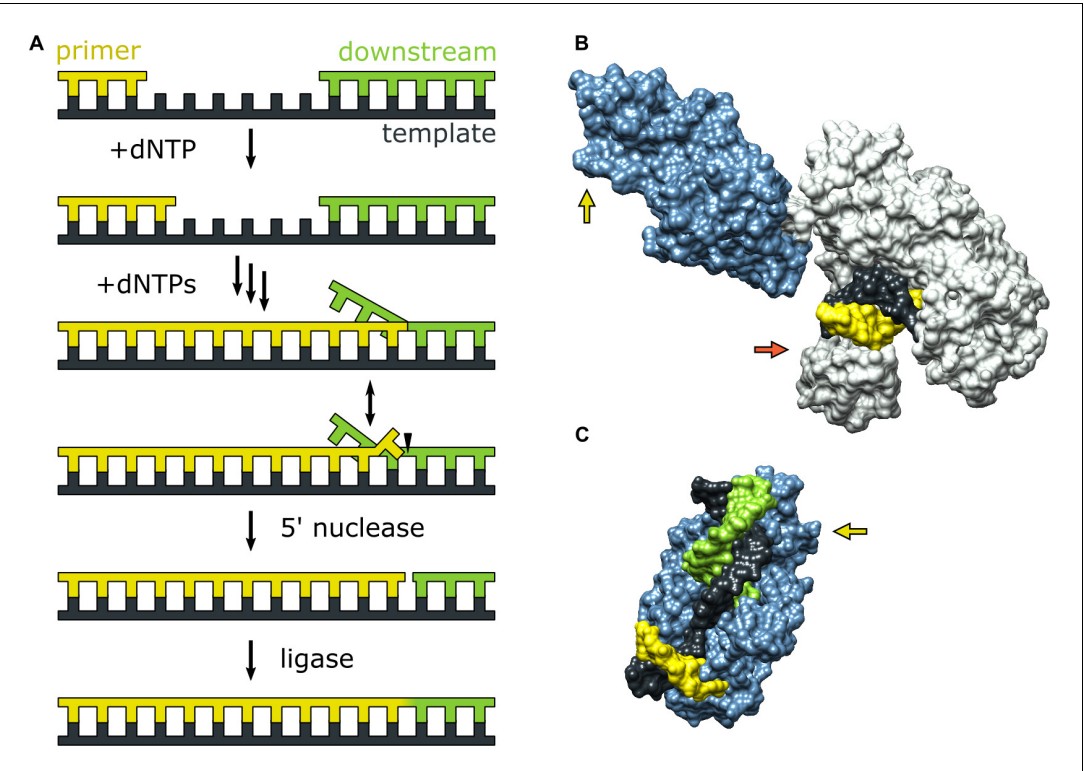

**Figure 1.** Activities of Pol I and three-dimensional structures of Pol I homologs. (**A**) Processes catalyzed by Pol I during lagging strand DNA replication or base excision repair. A growing primer strand is extended through repeated cycles of nucleotide incorporation, resulting in displacement of a downstream strand (RNA in the case of Okazaki fragment maturation or DNA in the case of base excision repair). The resulting substrate rearranges to a double flap structure that is cleaved by the 5' nuclease activity of Pol I (site indicated by a half arrow head). The final ligation step is performed by a DNA ligase (not shown). (**B**) Crystal structure of the Pol I homolog *Taq* polymerase with DNA primer/template bound in the *pol* domain (PDB ID: 1TAU). The polymerase core is shown in gray and the *5' nuc* domain is shown in light blue. (**C**) Crystal structure of human FEN1 bound to a DNA substrate (PDB: 3Q8M). In both B and C, the DNA strands are colored as in A and the green and red arrows denote the approximate locations of the sites for donor (scheme 2) or acceptor (scheme 1) labeling, respectively.

can generate long downstream 5' flaps that are difficult to cleave (*Reha-Krantz, 2010*), while excessive *5' nuc* activity can result in extended regions of single-stranded DNA that are prone to breakage (*Zheng et al., 2005*). Either outcome is deleterious to genomic integrity and cellular survival. A previous biochemical study suggests that 5' flap cleavage is performed by the same Pol I molecule that extended the primer (*Xu et al., 2000*), indicating a high degree of functional coordination. However, the structural and physical basis for this coordination is unknown, since crystal structures of Pol I engaged with DNA substrates via the *pol* or *5' nuc* domains have not been reported to date. However, the structure of the Pol I homolog *Taq* polymerase bound to DNA (*Eom et al., 1996*) reveals that the primer 3' terminus is located within the palm region of the *pol* domain, while the *5' nuc* domain is extended away from the enzyme core (*Figure 1B*). Moreover, the structure of human FEN1, which is homologous to the *5' nuc* domain of Pol I, in complex with a DNA substrate provides a model for how the *5' nuc* domain of Pol I engages a DNA substrate (*Tsutakawa et al., 2011*). The structure shows a single unpaired base at the primer 3' terminus inserted into a pocket of FEN1 and a region of extended contacts between the protein and the sugar-phosphate backbones of both strands of the downstream DNA (*Figure 1C*). These structures of homologous enzymes suggest that the *pol* and *5' nuc* binding modes of Pol I must differ significantly, raising questions about how the enzyme switches from one mode of activity to another.

Theoretical modeling of the transition from *pol* to *5' nuc* activity in Pol I, starting from the extended enzyme conformation shown in *Figure 1B*, suggests that the *5' nuc* domain flips by ~180°

and adopts a position above the fingers and thumb subdomains within the enzyme core (*Xie and Sayers, 2011*), facilitating interactions with the primer terminal base and a downstream DNA flap. Additionally, it is likely that the primer terminal base must detach from the *pol* domain in order to insert into the *5' nuc* domain, as seen in the FEN1-DNA structure (*Figure 1C*). However, these hypothetical movements of the DNA substrate and *5' nuc* domain have not been observed experimentally.

Here, we use single-molecule Förster resonance energy transfer (smFRET) microscopy to investigate the physical basis of functional coordination in Pol I and to probe large-scale conformational changes of the enzyme-DNA complex. The smFRET method has been previously applied to Klenow fragment (KF), which is derived from Pol I by removal of the *5' nuc* domain. These studies have monitored DNA synthesis by KF (*Christian et al., 2009*), detected nucleotide-induced conformational transitions within KF (*Santoso et al., 2010*; *Berezhna et al., 2012*; *Hohlbein et al., 2013*), monitored a DNA primer switching between the *pol* and *exo* sites of KF (*Lamichhane et al., 2013*) and established a structural model of a DNA substrate bound to KF (*Craggs et al., 2019*). However, smFRET has not yet been applied to full-length Pol I, or any other DNA polymerase containing a *5' nuc* activity.

In this study, we have developed an smFRET system to identify separate subpopulations of DNA engaging the *pol* and *5' nuc* domains of Pol I, revealing how the fractional populations and dwell times of each species vary according to the nature of the DNA substrate. We have also implemented a complementary smFRET system to probe the location of the flexibly tethered *5' nuc* domain, revealing that this domain undergoes a large positional shift in order to interact with a downstream DNA strand. Importantly, using both smFRET systems, we demonstrate that DNA substrates can switch between the *pol* and *5' nuc* domains during a single encounter with Pol I. Altogether, the information from the smFRET experiments reported here provides new insights into the physical mechanisms and enzyme conformational changes that underlie functional coordination in Pol I and are likely relevant to other multifunctional DNA polymerases with spatially separated active sites.

## Results

### Experimental design

We designed a series of model DNA substrates containing elements expected to engage the *5' nuc* domain of Pol I. The substrates are shown in schematic form in *Figure 2* and the complete sequences of the constituent oligonucleotides are listed in Appendix 1. One substrate contains a single-stranded 5' flap on the downstream strand, referred to as downstream flap DNA (*Figure 2A*). Another substrate contains the downstream flap and a single unpaired 3' terminal base on the primer strand (referred to as double flap DNA, *Figure 2B*). A third substrate can potentially exist as a mixture of downstream flap and double flap forms (referred to as mixed flap DNA, *Figure 2C*). We also examined DNA substrates containing a nick or gap of various sizes (*Figure 2D*) or lacking a downstream strand entirely (*Figure 2E*). In all cases, the template contains a $dT_{10}$ spacer and biotin group at the 3' end for surface attachment. Individual encounters between Pol I, present in solution, and the surface-immobilized DNA substrates were visualized by smFRET microscopy. Two different FRET labeling schemes were employed (*Figure 2F* and *Figure 2G*), as described in more detail in the following sections.

### Movement of DNA between *pol* and *5' nuc* domains of Pol I

The first FRET system was designed to probe the location of the DNA substrates relative to the enzyme core. A base located within the primer strand was labelled with an Alexa Fluor 488 (A488) FRET donor and Pol I was labeled at position 550 in the thumb region with a complementary Alexa Fluor 594 (A594) FRET acceptor (*Figure 3A* and *Appendix 1—table 1*). To achieve site-specific labeling of Pol I, the two native cysteines were removed via C262S and C907S mutations, and a single cysteine was introduced at the desired labeling site via a K550C mutation. The Pol I construct also contained D424A and D116A mutations to eliminate 3'–5' exonuclease and 5' nuclease activities, respectively (*Derbyshire et al., 1991*; *Xu et al., 2001*). Pol I was expressed and purified as described in the Materials and methods section and each step of the purification process was monitored by PAGE (Appendix 2). Overall $K_d$ values for binding of Pol I to the flap-containing DNA

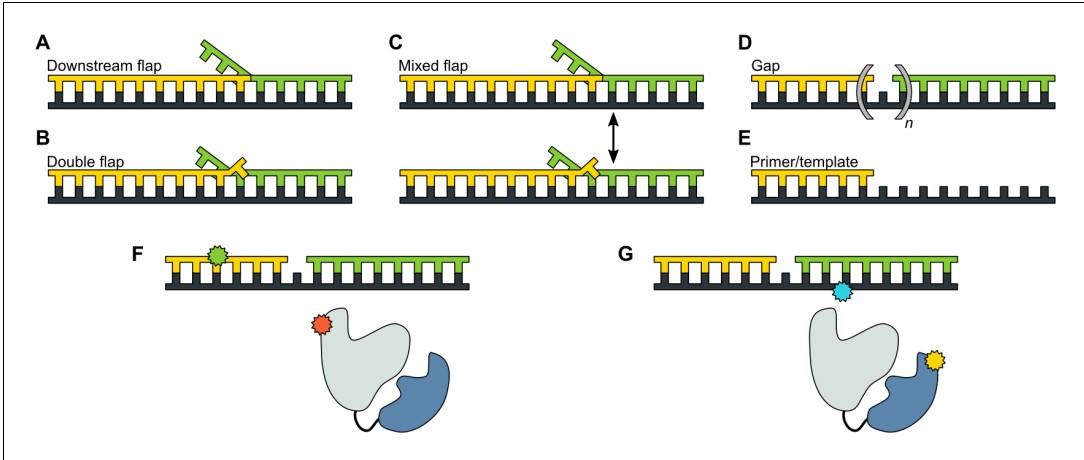

**Figure 2.** DNA substrates used in this study. (**A**) Substrate containing a 5' flap on the downstream strand (designated downstream flap DNA). (**B**) Substrate containing the same 5' flap as in A, plus a single unpaired base at the primer 3' terminus (designated double flap DNA). Because of the base sequences of the strands (Appendix 1), the structures shown in A and B are 'locked in'. (**C**) Substrate that can exist as a mixture of the structures shown in A and B (designated mixed flap DNA). (**D**) Substrates containing a nick ($n = 0$) or gaps of various size ($n = 1$–4). (**E**). Primer/template substrate. (**F**) Schematic illustration of donor (green) and acceptor (red) labeling sites for the first FRET scheme. Pol I is depicted in cartoon form, with the core colored grey and the 5' *nuc* domain colored blue. (**G**) Schematic illustration of donor (yellow) and acceptor (blue) labeling sites for the second FRET scheme.

substrates (*Appendix 3—table 1*) are similar to values reported for KF interacting with primer/template substrates (*Christian et al., 2009*; *Kuchta et al., 1988*; *Polesky et al., 1990*), indicating that the mutations within our Pol I construct do not disrupt the DNA-binding ability of the protein.

A representative set of donor, acceptor and FRET efficiency trajectories depicting a series of encounters between Pol I and immobilized DNA (mixed flap in this example) is shown in *Figure 3B*. During each encounter, the donor intensity abruptly drops, and an acceptor signal appears at the same instant, reflecting binding of Pol I to the DNA, while at a later time point the acceptor signal disappears and the donor signal increases correspondingly, reflecting dissociation of Pol I from the DNA. The anti-correlation of the donor and acceptor signals confirms that the intensity changes are due to FRET. During the binding periods, the FRET efficiency alternates between two distinct levels at ~0.8 or ~0.6 efficiency. Additional examples of smFRET trajectories are presented in *Figure 3—figure supplement 1*. Multiple FRET trajectories for each DNA substrate interacting with Pol I were analyzed globally using a Hidden Markov model, confirming that two distinct bound states are sufficient to account for all data sets. The resulting composite FRET efficiency histograms for each state are shown in *Figure 3C*. Note that each histogram is accumulated during the global HMM analysis, not by Gaussian fitting of final envelopes, ensuring clean separation of the two states and accurate quantification of the state populations.

A state at 0.8 FRET efficiency was previously observed for DNA primer/templates interacting with KF (same donor and acceptor labeling sites as here) and was attributed to DNA engaging the *pol* domain (*Lamichhane et al., 2013*). Since KF is identical to the main core of Pol I, the 0.8 FRET state observed here is also attributed to DNA engaging the *pol* domain within Pol I (state P). To test whether the 0.6 FRET state arises from DNA binding to the *exo* domain of Pol I, we introduced a L361A mutation, which is known to disrupt DNA binding at the *exo* domain of KF (*Lamichhane et al., 2013*; *Lam et al., 1998*). The L361A mutation has little effect on the 0.6 FRET population for the mixed flap and downstream flap substrates (compare top and middle rows in *Figure 3C*), indicating that this state does not arise from binding of these DNAs to the *exo* domain. However, for the double-flap substrate, the presence of the L361A mutation decreased the population of the 0.6 FRET state by 15% (*Figure 3C*), suggesting that this state arises, in part, from binding of the DNA to the *exo* domain. This is not unexpected, because the primer terminal base is mispaired with the template base in this DNA (*Appendix 1—table 1*). Consistent with this, a primer/template substrate containing the same terminal mismatch (*Appendix 1—table 1*) exhibited a state

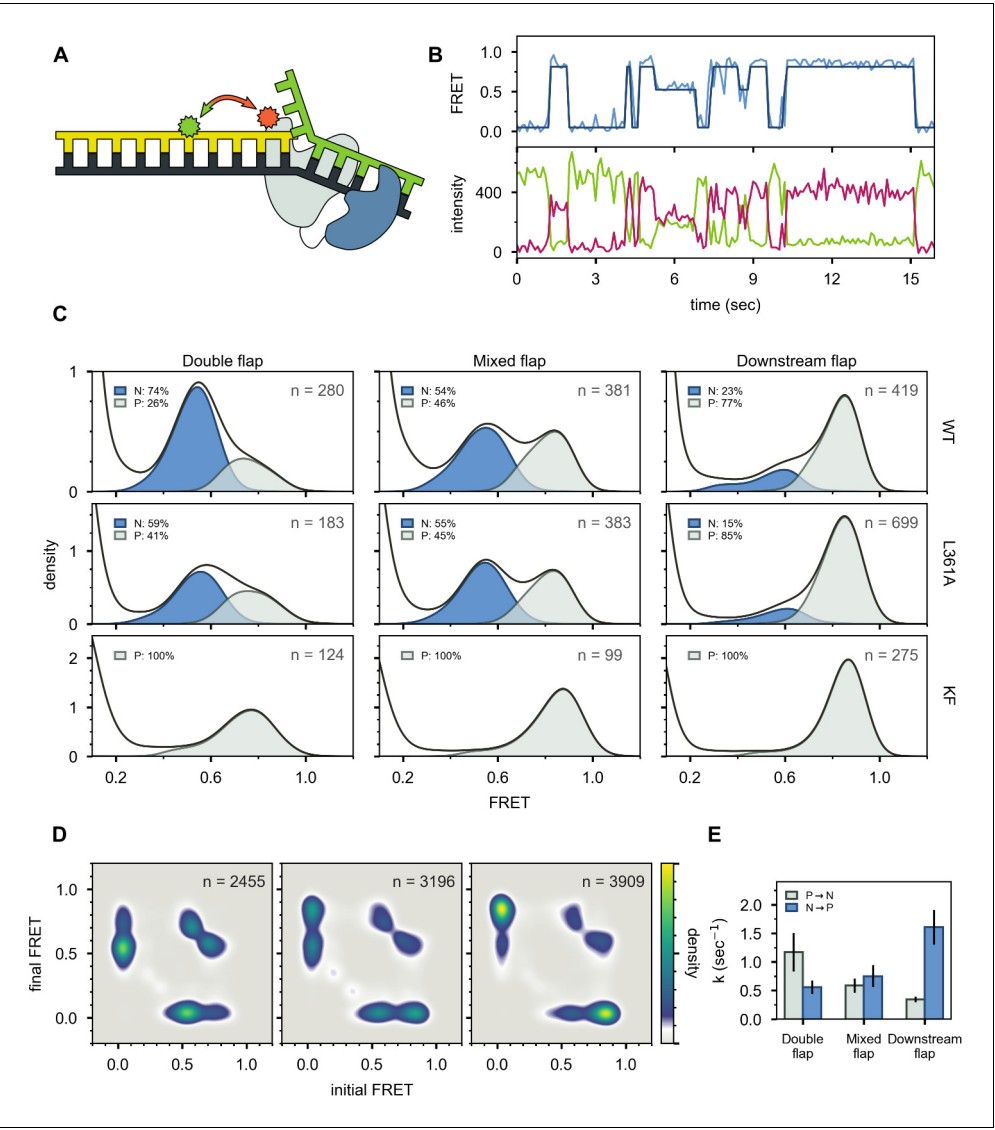

**Figure 3.** Probing the location of DNA substrates within Pol I. (**A**) Schematic representation of the donor (green, attached to primer strand) and acceptor (magenta, attached to thumb region of Pol I) labeling sites. Pol I is depicted in cartoon form, with the core colored gray and the 5' *nuc* domain colored blue. (**B**) Representative smFRET trajectory (blue) and donor (green) and acceptor (magenta) emission trajectories, for Pol I interacting with mixed flap substrate. The bold line is the idealized state path determined from Hidden Markov modeling. (**C**) Composite FRET efficiency histograms for states P and N, compiled from *n* individual FRET trajectories (*n* value indicated in each plot), for various combinations of DNA substrate and protein. The proteins, from top to bottom, are WT Pol I, Pol I L361A, and KF L361A. The corresponding populations of states P and N are indicated. (**D**) Transition density plots for Pol I interacting with flap-containing DNA substrates, compiled from a total of *n* transitions. From left to right: double flap DNA, mixed flap DNA and downstream flap DNA. (**E**) Rate constants for intramolecular transfer of DNA substrates from *pol* domain to 5' *nuc* domain (P→N) or from 5' *nuc* domain to *pol* domain (N→P).

The online version of this article includes the following figure supplement(s) for figure 3:

**Figure supplement 1.** Additional examples of smFRET trajectories (blue) and corresponding donor intensity (green) and acceptor intensity (red).

**Figure supplement 2.** Interaction of Pol I with mismatched primer/template.

**Figure supplement 3.** Histogram of emission intensities resulting from direct excitation of A594 in Pol I bound to downstream flap DNA, compiled from 102 individual emission trajectories.

**Figure supplement 4.** Dwell time histograms for overall decay of state P (left column) or state N (right column) for Pol I interacting with flap-containing DNA substrates, compiled from a total of *n* transitions.

at 0.6 FRET efficiency when bound to Pol I, and this state was completely eliminated with a L361A mutation within Pol I *Figure 3—figure supplement 2*, confirming that mismatched DNA can engage the *exo* domain within Pol I. We also examined KF, which lacks the *5' nuc* domain entirely. The KF construct was also labeled at position 550 with A594 and contained an L361A mutation. Notably, the 0.6 FRET population is not observed for any of the DNA substrates in the presence of KF (*Figure 3C*, bottom row). Together, these observations indicate that the 0.6 FRET state arises primarily from DNA engaging the *5' nuc* domain of Pol I (state N). In the case of the double flap substrate, the 0.6 FRET state also reflects a small (15%) population engaging the *exo* domain of Pol I (state E). Interestingly, this population is significantly less than observed for the primer/template containing the same terminal mismatch (53%, *Figure 3—figure supplement 2*), indicating that the presence of the downstream strand in the double flap substrate inhibits binding of DNA to the *exo* domain and/or favors binding to the *5' nuc* domain.

The lower FRET efficiency of state N could be due to movement of the upstream duplex (where the donor is located) as the primer terminus detaches from the *pol* domain and moves to the *5' nuc* domain, tilting of the protein helix to which the acceptor is attached, or changes in local environment of donor and/or acceptor that alter the Förster radius. Photophysical control experiments indicate that the Förster radius in state N is similar to that in state P (Appendix 4), indicating that the lower FRET efficiency of state N is due to physical movement of the DNA substrate and/or thumb subdomain. However, since the thumb subdomain is a rigid structural element within Pol I and KF, it is likely that the FRET change is actually due to movement of the DNA substrate. The change in FRET efficiency from 0.8 to 0.6 corresponds to a lengthening of the donor-acceptor distance of ~7 Å (Appendix 4). This movement of the upstream DNA duplex is similar for states N and E, indicating that the DNA must detach from the *pol* domain in order for the primer 3' terminus to reach either the *5' nuc* domain or the *exo* domain.

State N is highly populated for the double-flap substrate (59%, after accounting for the population of state E, *Figure 3C*, top left) and mixed-flap substrate (54%, *Figure 3C*, top middle). Both DNAs contain or can form double-flap structures that are the preferred substrates for the *5' nuc* activity of Pol I (*Xu et al., 2001*). In contrast, state N is least populated with the downstream flap substrate (23%, *Figure 3C*, top right).

Two-dimensional transition density plots (TDPs) (*McKinney et al., 2006*) were constructed from multiple FRET trajectories to reveal the connectivity of the various FRET states (*Figure 3D*). The two peaks evident on the y-axis reflect binding of Pol I to the immobilized DNA, using either the *pol* or *5' nuc* domains, while the peaks on the x-axis reflect the corresponding dissociation transitions. These results imply that Pol I can engage DNA via one domain (*pol* or *5' nuc*), dissociate into bulk solution, and subsequently rebind using the other domain. An example of such an event sequence (dissociate from state N and rebind in state P) is evident in the representative smFRET trajectory shown in *Figure 3B* (middle portion, from ~6s to ~8s). Dissociation and rebinding provide one pathway for transfer of DNA substrates between *pol* and *5' nuc* domains (intermolecular transfer). In addition, we observe frequent transitions between states P and N that do not show a measurable pause in a zero-FRET state (examples are shown in *Figure 3B*, between ~8s and ~10s and in *Figure 3—figure supplement 1*). These direct transitions give rise to prominent cross peaks between 0.6 and 0.8 FRET states in the TDPs (*Figure 3D*).

There are two models that account for the cross peaks evident in the TDPs. First, a single Pol I molecule bound to DNA via the *pol* domain could spontaneously shift to engage the DNA via the *5' nuc* domain without escaping into bulk solution (intramolecular transfer). Alternatively, two Pol I molecules could bind to a single DNA substrate, one engaging the DNA via the *pol* domain and the other engaging the DNA via the *5' nuc* domain. Dissociation of the first Pol I molecule would give rise to a transition from 0.8 to 0.6 FRET efficiency. To distinguish these possibilities, we evaluated the stoichiometry of Pol I on each of the DNA substrates. To do so, we excited the A594 acceptor directly and recorded the resulting A594 emission over time, under the same conditions used for the smFRET experiments. A histogram of A594 emission intensity, compiled from 102 individual Pol I molecules interacting with immobilized DNA (downstream flap in this example), reveals a peak at ~350 a.u. (*Figure 3—figure supplement 3*), corresponding to a single Pol I molecule bound to DNA (the peak at zero intensity is due to periods in which Pol I is not bound to the DNA). Notably, there is no indication of a peak or shoulder at ~700 a.u., corresponding to two bound Pol I molecules. Similar results were obtained for the other DNA substrates (not shown). We conclude that a single Pol I

molecule is bound to DNA under the conditions of our experiments, ruling out the second scenario described above.

Rate constants for overall decay of states P and N were determined for each DNA substrate by dwell time analysis (*Figure 3—figure supplement 4*). These observed rate constants were separated into the microscopic rate constants for various state-to-state transitions, according to *Equations 3 and 4* in Materials and methods and using the transition frequencies listed in *Appendix 3—table 2*. Transfer from the *pol* domain to 5' *nuc* domain is fastest for the double flap substrate and slowest for the downstream flap substrate (*Figure 3E*, *Appendix 3—table 3*). In contrast, the rates of dissociation from the *pol* domain are similar for each substrate (*Appendix 3—table 3*). Hence, transfer from the *pol* domain to the 5' *nuc* domain is kinetically distinct from overall dissociation from the *pol* domain, emphasizing the intramolecular nature of the process.

Kinetic analysis also reveals significant variations in the rates of intramolecular transfer from the 5' *nuc* domain to the *pol* domain among the various DNA substrates (*Figure 3E* and *Appendix 3—table 3*). The downstream flap substrate exhibits the fastest transfer. In contrast, transfer of the double flap substrate is much slower, showing the importance of the unpaired primer base for stable engagement with the 5' *nuc* domain. The structure of FEN1 (homologous to the 5' *nuc* domain of Pol I) with DNA reveals that the unpaired primer base is sequestered in a binding pocket, making contacts with a network of surrounding protein residues (*Tsutakawa et al., 2011*). A similar arrangement in Pol I would account for the prolonged residence time of DNA in state N and the slow return to state P.

## Role of downstream strand and 5' flap for binding to the 5' *nuc* domain

To investigate whether the presence of a downstream strand is required to engage the 5' *nuc* domain, we examined a DNA substrate containing only primer and template strands (*Figure 2E* and *Appendix 1—table 1*). This substrate exhibits a single FRET state at ~0.8 efficiency, corresponding to state P, while interacting with Pol I (*Figure 4A and B*, bottom panels). Likewise, no cross peaks are observed in the TDP (*Figure 4C*). State N is absent in this case, showing that the downstream DNA must be in duplex form to engage the 5' *nuc* domain and promote movement of the upstream DNA out of the *pol* domain.

To test whether the presence of a 5' flap on the downstream strand is also required to engage the 5' *nuc* domain, we examined DNA substrates containing a nick or single-stranded gaps of various sizes (*Figure 2D* and *Appendix 1—table 1*). These substrates reveal reversible transitions between ~0.6 and ~0.8 FRET states when interacting with Pol I (*Figure 4A*, upper three panels), similar to the substrates containing 5' flaps. Moreover, the hallmarks of the 0.6 FRET state are indicative of DNA engaging the 5' *nuc* domain (state N, *Figure 4—figure supplement 1*). The populations of state N for the nicked and single-nucleotide gap substrates (26% and 23%, respectively, *Figure 4B*) are similar to the downstream flap substrate (23%, *Figure 3C*), indicating that the presence of a downstream strand is sufficient to engage the 5' *nuc* domain, regardless of whether that strand contains a 5' flap or not. Moreover, the 5' *nuc* domain can engage a downstream strand even when it is separated from the primer strand by a 4nt gap (*Figure 4A and B*), which is likely a consequence of the flexible 16 aa linker tethering the 5' *nuc* domain to the body of the enzyme. The flexibility of the single-stranded gap region could also facilitate docking of the 5' *nuc* domain with the downstream DNA strand. The presence of prominent cross peaks in the TDP shows that a gapped DNA substrate can transfer reversibly between *pol* and 5' *nuc* domains during a single encounter with Pol I (*Figure 4C*). Rate constants for state-to-state transitions were determined by combining overall decay rates of states P and N (*Figure 4—figure supplement 2*) and transition frequencies (*Appendix 3—table 4*), as described above. Transfer from the *pol* domain to the 5' *nuc* domain becomes progressively slower as the gap size increases (*Figure 4D* and *Appendix 3—table 5*). Return of all DNAs from the 5' *nuc* domain to the *pol* domain is more rapid (*Figure 4D* and *Appendix 3—table 5*). Overall $K_d$ values for binding of Pol I to primer/template, nicked and gapped DNA substrates are listed in *Appendix 3—table 6*. Interestingly, dissociation of Pol I from nicked DNA (from either state P or state N) is significantly faster than from any of the gapped DNAs (*Appendix 3—table 5*), suggesting that Pol I is kinetically tuned to dissociate from a nicked substrate, allowing a ligase enzyme to access the DNA.

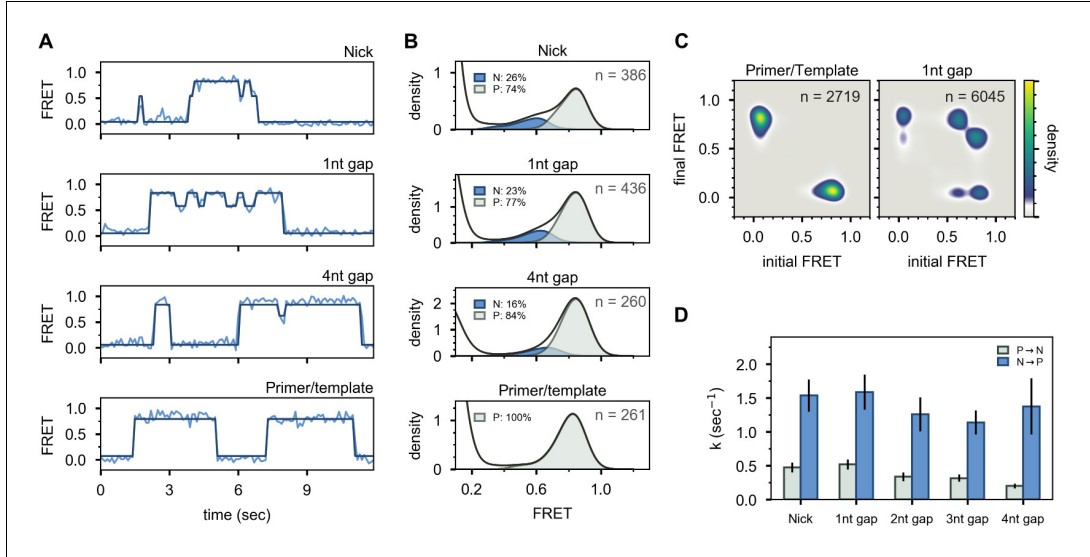

**Figure 4.** Interaction of Pol I with primer/template DNA or DNA substrates containing a nick or gap. (**A**) Representative smFRET trajectories for DNA substrates interacting with Pol I, as indicated. Schematic representations of DNA substrates are shown in *Figure 2*. Bold lines are idealized state paths determined from Hidden Markov modeling. (**B**) Composite FRET efficiency histograms for states P and N compiled from *n* individual FRET trajectories, for various DNA substrates interacting with Pol I, as indicated. The corresponding populations of states P and N are indicated. (**C**) Transition density plots for Pol I interacting with various DNA substrates compiled from a total of *n* transitions, as indicated. (**D**) Rate constants for intramolecular transfer of various DNA substrates between *pol* domain and *5' nuc* domain (P→N, grey) or between *5' nuc* domain and *pol* domain (N→P, blue).

The online version of this article includes the following figure supplement(s) for figure 4:

**Figure supplement 1.** Histograms and TDPs of Pol I L361A and KF interacting with 1nt gap DNA.

**Figure supplement 2.** Dwell time histograms for overall decay of state P (left column) or state N (right column) for Pol I interacting with nick- or gap-containing DNA substrates, compiled from a total of *n* transitions.

## Movement of the *5' nuc* domain during docking with downstream DNA

The second FRET system is designed to probe the proximity of the *5' nuc* domain to the downstream portion of the DNA substrates. In this case, an A488 donor was attached to a cysteine residue introduced at position 213 in the *5' nuc* domain of Pol I (via a T213C mutation) and an A594 acceptor was attached to a base in the downstream portion of the template strand (*Figure 5A* and *Appendix 1—table 2*). The DNA substrates were otherwise analogous to those used in scheme 1. The A594 acceptor was placed in the template, rather than downstream strand, to avoid any possible steric interference with binding of Pol I. The Pol I construct also contained C262S, C907S, D424A, and D116A mutations, as described above. A representative set of donor, acceptor and FRET efficiency trajectories for the mixed flap DNA substrate is shown in *Figure 5B*. For this labeling scheme, there is no signal from either donor or acceptor during the periods when Pol I is not present on the DNA. Upon binding of Pol I, both donor and acceptor signals appear simultaneously and their relative magnitudes reflect the FRET efficiency during the encounter. Upon dissociation of Pol I, both signals disappear at the same instant (*Figure 5B*).

Multiple FRET trajectories for each DNA substrate (containing 5' flap) interacting with Pol I were analyzed globally using a Hidden Markov model, showing that two distinct states were sufficient to account for all data sets. The resulting FRET efficiency histograms for each state are centred at ~0.8 and ~0.6 efficiency (*Figure 5C*). Notably, the fractional populations of the high-FRET and mid-FRET peaks are similar to the fractional populations of states N and P, respectively, observed with the first labeling scheme (compare *Figure 5C* and *Figure 3C*). This is true across all three substrates containing 5' flaps, which partition differently between states N and P. Accordingly, the high-FRET and mid-FRET species observed with the present labeling scheme are assigned to states N and P, respectively. The high FRET efficiency of state N indicates that the donor and acceptor sites are relatively

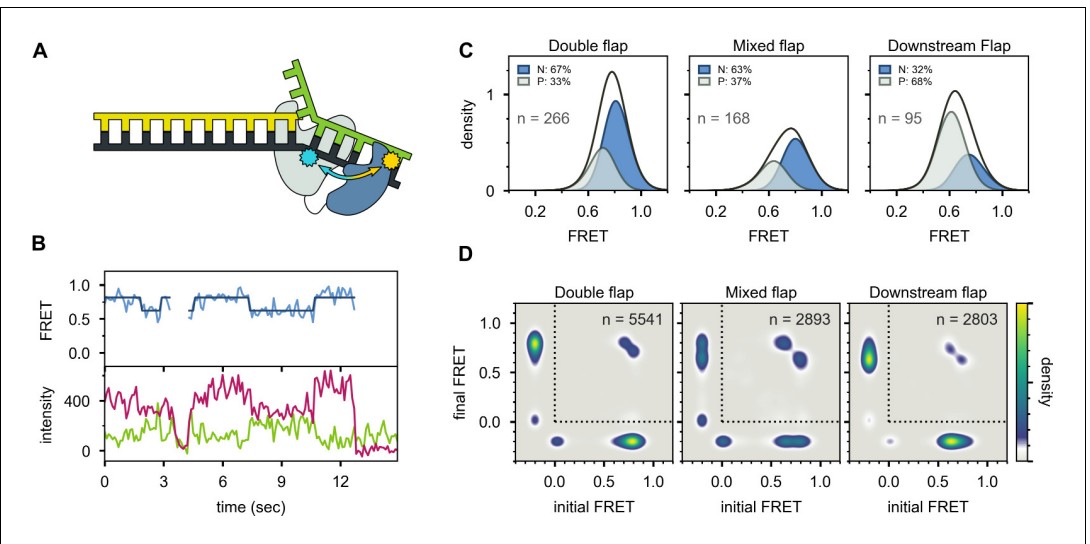

**Figure 5.** Probing the location of the 5' *nuc* domain within Pol I bound to flap substrates. (**A**) Schematic representation of donor (yellow, attached to 5' *nuc* domain) and acceptor (cyan, attached to downstream template strand) labeling sites. (**B**) Representative set of donor, acceptor and FRET trajectories for mixed flap DNA substrate. The bold lines are idealized state paths from Hidden Markov modeling. (**C**) FRET efficiency histograms compiled from *n* individual FRET trajectories. (**D**) Transition density plots compiled from a total of *n* transitions. Since the FRET efficiency is not defined during periods when Pol I is not bound to DNA, the FRET efficiency is set to −0.2. The quadrant enclosed by the dotted lines corresponds to periods during which Pol I is bound to DNA.

close in space (donor-acceptor distance ~40 Å), confirming that state N arises from engagement of the 5' *nuc* domain with the downstream DNA. The lower FRET efficiency of state P indicates that the 5' *nuc* domain is somewhat further from the downstream DNA when the primer 3' terminus occupies the *pol* domain (donor-acceptor distance ~49 Å). Direct transitions between the mid-FRET and high-FRET states are observed in individual FRET trajectories (*Figure 5B*) and in composite TDPs (*Figure 5D*), indicating that Pol I can switch between states P and N without dissociation. This conclusion is consistent with the observations from the first labeling scheme (*Figure 3D*).

In striking contrast, the primer/template substrate mostly populates a single FRET state with a much lower efficiency of ~0.3 (*Figure 6*), indicating that the 5' *nuc* domain is distant from the downstream template (donor-acceptor distance ~60 Å). We designate this state P' (see Discussion). The FRET histogram also reveals a barely detectable shoulder at ~0.6 efficiency, corresponding to state P (*Figure 6C*). There are very few transitions between states P' and P, as indicated by the absence of cross peaks in the TDP (*Figure 6D*). Overall, these observations are consistent with the first

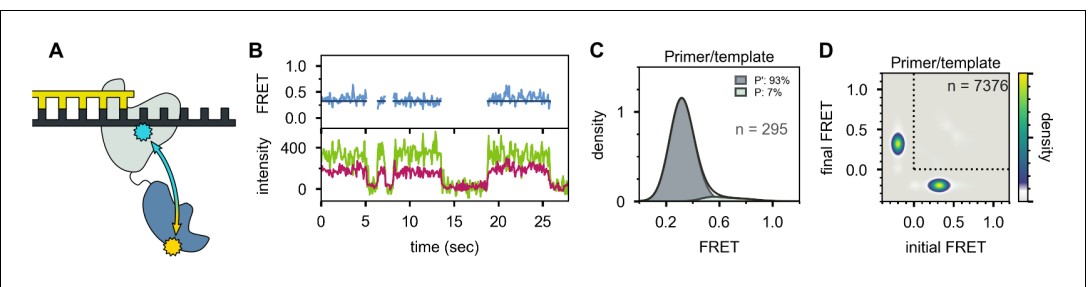

**Figure 6.** Location of the 5' *nuc* domain within Pol I bound to primer/template substrate. (**A**) Schematic representation of donor (yellow) and acceptor (cyan) labeling sites. (**B**) Representative set of donor, acceptor and FRET trajectories. (**C**) FRET efficiency histograms compiled from *n* individual FRET trajectories. (**D**) Transition density plot compiled from a total of *n* transitions. Same presentation as *Figure 5D*.

labeling scheme, which indicates that the downstream DNA must be present in duplex form in order to engage the 5' *nuc* domain (*Figure 4B*).

## Discussion

The 5' *nuc* domain of Pol I fulfils a key function during DNA replication and repair, cleaving 5' flaps that arise from strand displacement synthesis (*Figure 1A*). The physical and structural mechanisms that underlie the coordination between the *pol* and 5' *nuc* activities of Pol I are not well understood, owing to technical challenges in studying this enzyme. The 5' *nuc* domain is tethered to the enzyme core by an unstructured 16 aa peptide linker, which may allow this domain to adopt a range of positions. This intrinsic flexibility has likely impeded efforts to crystallize Pol I and determine a high-resolution crystal structure. More fundamentally, the potential ability of the 5' *nuc* domain to exchange between different positions within a Pol I-DNA complex could play a role in orchestrating the transition from one mode of enzymatic activity to another.

Here, we have developed two complementary smFRET systems to visualize Pol I spontaneously exchanging between different DNA binding modes and to probe the location of the 5' *nuc* domain in each case. Using the first labeling scheme, we have resolved two distinct FRET states and shown that they correspond to DNA engaging the *pol* domain (state P) or engaging the 5' *nuc* domain (state N). Although we have previously observed the *pol*-binding mode in analogous smFRET studies of KF (*Lamichhane et al., 2013*), the present spectroscopic observations of the 5' *nuc* binding mode in full-length Pol I have not been reported before. Using the second labeling scheme, we have confirmed that state N arises from docking of the 5' *nuc* domain with the downstream DNA. Notably, the fractional populations of states P and N determined with each labeling scheme are in agreement (*Figure 3C* and *Figure 5C*), establishing a consistent description of the Pol I-DNA complexes under study. In contrast, if a downstream strand is not present, the 5' *nuc* domain adopts a different position, extended way from the DNA substrate (*Figure 6*), underscoring the mobility of this independent and flexibly tethered protein domain.

Although structures of full-length Pol I with DNA substrates are not available, our spectroscopic data on states P and N are consistent with details revealed in the structures of homologous enzymes bound to DNA substrates. The crystal structure of *Taq* polymerase with a DNA primer/template engaging the *pol* domain shows the 5' *nuc* domain extended away from the enzyme core (*Figure 1B*), consistent with our smFRET observations using scheme 2 (*Figure 6*). Moreover, comparison of the co-crystal structures of *Taq* polymerase (*Figure 1B*) and FEN1 (*Figure 1C*) imply that the primer 3' terminal base is located in distinct locations in each case. In the context of Pol I, these structures suggest the primer terminal base detaches from the *pol* domain and inserts into the 5' *nuc* domain, which likely requires some movement of the entire DNA substrate. This is consistent with our observation that the DNA duplex undergoes a ~7 Å displacement between states P and N. Moreover, the observation that the primer terminal base is unpaired in the FEN1 structure is consistent with our spectroscopic observations that state N is most highly populated with the double-flap DNA substrate (*Figure 3C* and *Figure 5C*). The FEN1 structure also shows a region of extended contacts between the protein and the sugar-phosphate backbones of both strands of the downstream DNA (*Tsutakawa et al., 2011*), consistent with our observations that the downstream DNA must be in duplex form to engage the 5' *nuc* domain (*Figure 4B*) and that a downstream strand engages the 5' *nuc* domain of Pol I even when separated from the primer terminus by a 4nt gap (*Figure 4B*).

An important finding from our smFRET study is that DNA substrates can transfer reversibly between *pol* and 5' *nuc* domains during a single encounter with Pol I. These intramolecular transitions are readily observable in individual FRET trajectories (*Figure 3B* and *Figure 5B*) and in composite transition density plots (*Figure 3D* and *Figure 5D*), obtained using either labeling scheme. Our results explain previous biochemical observations indicating that the same Pol I molecule that extends the primer terminus can also cleave the resulting downstream flap (*Xu et al., 2000*). An intramolecular transfer pathway is also employed during *pol* to *exo* switching in KF (*Lamichhane et al., 2013*; *Joyce, 1989*).

Our results demonstrate that the double flap substrate can also engage the *exo* domain of Pol I (state E). However, the fractional population of state E is significantly smaller (15%) than observed for a corresponding primer/template containing the same terminal mispair (53%), suggesting that the presence of the downstream strand in the double flap substrate favors binding to the 5' *nuc*

domain and/or suppresses binding to the *exo* domain. This observation could imply that 5' flap cleavage takes precedence over exonucleolytic proofreading, although our observations performed under equilibrium conditions may not recapitulate the kinetic competition between these two pathways. Interestingly, the FRET efficiencies for states N and E in scheme one are similar. Apparently, the upstream DNA duplex undergoes a similar displacement as the primer terminus detaches from the *pol* domain and moves to either the 5' *nuc* domain or the *exo* domain.

Taken together, the results from the two FRET systems suggest that complexes of Pol I with DNA can adopt three distinct configurations, summarized schematically in *Figure 7*. Complex P' is formed exclusively with a primer/template substrate: the primer 3' terminus is bound in the *pol* domain and the 5' *nuc* domain is extended away from the enzyme core. The prime symbol is to distinguish this species from complex P, which forms with DNA substrates containing a downstream strand: the primer 3' terminus is still located in the *pol* domain, but the 5' *nuc* domain is in proximity to the downstream strand. Complexes P and P' are indistinguishable using the first labeling scheme, but are clearly resolved with the second scheme, highlighting the importance of utilizing multiple donor and acceptor sites to detect all species present. In complex N, the primer terminus has shifted from the *pol* domain to the 5' *nuc* domain and that domain is even closer to the downstream strand. Our results show that complexes P and N can freely interconvert, with the distribution of the two species being determined by the nature of the DNA substrate. Complex P is the dominant species for the downstream flap and 1nt gap substrates, with transient excursions to complex N. In contrast, the double flap substrate is biased towards complex N, consistent with the known substrate preference of the 5' *nuc* activity of Pol I (*Xu et al., 2001*). Theoretical modelling also suggests that the 5' *nuc* domain can adopt different positions within a Pol I-DNA complex (*Xie and Sayers, 2011*).

The complexes in *Figure 7* likely correspond to snapshots during strand displacement synthesis and the transition from *pol* to 5' *nuc* activity (*Figure 1A*). Complex P' corresponds to an early stage,

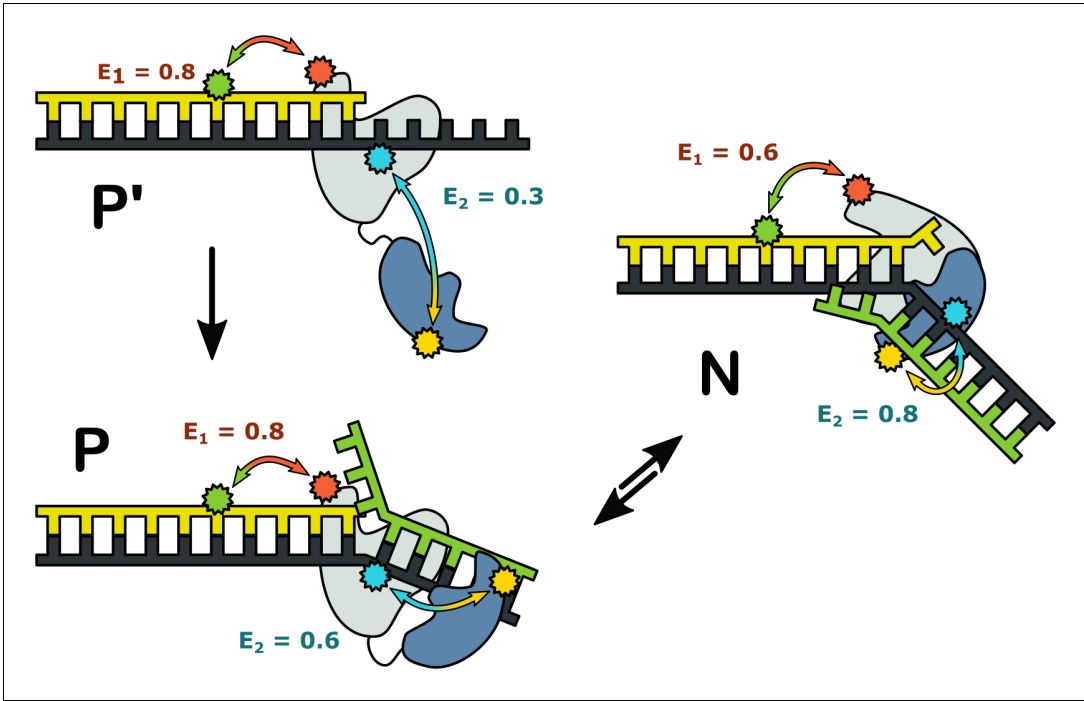

**Figure 7.** Possible configurations of Pol I-DNA complexes. Complex P': in the absence of a nearby downstream strand, the primer 3' terminus resides in the *pol* domain within the enzyme core (grey) and the 5' *nuc* domain (blue) is extended away from the core. The donor and acceptor probes for the first labeling scheme are green and magenta, respectively, while the donor and acceptor for the second labeling scheme are yellow and blue. The FRET efficiencies for the first and second labeling schemes are denoted $E_1$ and $E_2$, respectively. Complex P: The primer 3' terminus resides in the *pol* domain and the 5' *nuc* domain is located in proximity to a nearby downstream strand. Complex N: The 5' *nuc* domain is docked with the downstream strand and the DNA primer 3' terminus resides within the same domain.

when the nascent primer strand is distant from a downstream strand. Complex P corresponds to a later stage, in which the growing primer has displaced the downstream strand but the primer 3' terminus is still located in the *pol* domain. Complex N likely represents the next step in the pathway, in which the primer 3' terminus has moved out of the *pol* domain and the *5' nuc* domain is poised to cleave the scissile phosphodiester within the downstream strand, although the actual cleavage step is blocked here by a D116A mutation.

The present study has been performed under equilibrium conditions, whereby the enzymatic activities of Pol I are disabled by mutations and nucleotide substrates are absent. Our results reveal the intrinsic conformational dynamics of Pol I that facilitate the transition from *pol* to *5' nuc* activity. Future smFRET studies with catalytically active Pol I and in the presence of nucleotides should provide further insights into the temporal order of events and the associated enzyme and DNA conformational changes during strand displacement synthesis and 5' flap processing.

DNA polymerases from many organisms also possess distinct enzymatic activities that must be carefully coordinated to ensure accurate and efficient DNA replication and repair. In DNA Pol III, the major replicative polymerase in *E. coli*, the *pol* and *exo* active sites are located in separate protein subunits within a multi-protein holoenzyme complex (*McHenry, 2011*; *Oakley, 2019*). A similar situation prevails in eukaryotic DNA polymerases (*Burgers and Kunkel, 2017*; *Raia et al., 2019*). Moreover, in eukaryotes 5' flap cleavage is performed by a separate enzyme, such as FEN1 (*Dehé and Gaillard, 2017*; *Stodola and Burgers, 2017*). Pol I is a relatively simple model of multi-functional DNA polymerases because it contains three distinct activities within a single polypeptide and does not require accessory proteins for proper function. Here, we have shown that DNA substrates can transfer reversibly between *pol* and *5' nuc* domains during a single encounter with Pol I. Moreover, we have shown that the flexibly tethered *5' nuc* domain adjusts its position to engage the downstream DNA strand. Intramolecular transfer of the DNA substrate, combined with protein conformational changes, orchestrates the transition from one mode of activity to another, without the need to dismantle and reassemble the enzyme-DNA complex. This paradigm for functional coordination is likely relevant to more complex multi-functional DNA polymerase holoenzyme complexes, in which the various active sites are also widely separated in space, albeit within separate protein subunits.

## Materials and methods

**Key resources table**

| Reagent type (species) or resource | Designation | Source or reference | Identifiers | Additional information |
|---|---|---|---|---|
| Strain, strain background (*Escherichia coli*) | CJ803 | Yale coli genetic stock center | | |
| Recombinant DNA | pXS67 | Yale coli genetic stock center | | |
| Commercial assay or kit | Q5 site-directed mutagenesis kit | New England Biolabs | #E0554 | |
| | Quik Change kit | Agilent | 200523 | |
| Chemical compound | Alexa Fluor 488 NHS ester | Thermo Fisher | A20000 | |
| | Alexa Fluor 488 $C_5$ maleimide | Thermo Fisher | A10254 | |
| | Alexa Fluor 594 NHS ester | Thermo Fisher | A20004 | |
| | Alexa Fluor 594 $C_5$ maleimide | Thermo Fisher | A10256 | |
| Software, algorithm | scikit_learn | other | | public domain |
| Software, algorithm | SciPy | other | | public domain |

## Oligonucleotides

All DNA oligonucleotides were purchased from IDT in purified form and used as delivered. Oligonucleotides containing an amino-modified dT were labeled with either Alexa Fluor 488 or Alexa Fluor 594 NHS ester (ThermoFisher Scientific) according to the manufacturer's protocol. Template strands contained a biotin group at the 3' end for immobilization on neutravidin-coated microscope slides. All oligonucleotide sequences and modifications are listed in *Appendix 1—table 1* and *Appendix 1—table 2*.

## Expression of Pol I derivatives

A Pol I expression vector was generated from the plasmid pXS67 (Yale Coli Genetic Stock Center, Strain CJ803) by site-directed mutagenesis using a QuikChange kit (Agilent) according to the manufacturer's protocol. This construct, referred to as wild-type (WT) in the text, also carried C262S and C907S mutations to remove the two native cysteines in Pol I, a D424A mutation to suppress 3'–5' exonuclease activity, a D116A mutation to suppress 5' nuclease activity and a 6× histidine tag attached to the C-terminus of the protein by a Gly-Pro-Gly linker. A Pol I construct containing an additional L361A mutation was generated from the WT construct by site-directed mutagenesis using a Q5 site-directed mutagenesis kit (NEB) according to the manufacturer's protocol. KF carrying K550C, D424A, and C907S mutations (and L361A, as indicated) was produced from a previously described plasmid (*Berezhna et al., 2012*; *Lamichhane et al., 2013*). Expression and purification of KF was carried out as previously described (*Berezhna et al., 2012*; *Joyce and Derbyshire, 1995*). Pol I was expressed in the same manner and purified as detailed below.

## Purification of Pol I derivatives

*E. coli* cells expressing Pol I were lysed by sonication in HisTrap Buffer A (50mM Tris-HCl, pH 7.5, 10 mM-mercaptoethanol, 10 mM imidazole) supplemented with 20 μM phenylmethylsulfonyl fluoride (PMSF). Cellular debris was removed by centrifugation at 7500×g for 15 min at 4° C, and the clarified cell extract was loaded onto a 5 *mL* HisTrap HP column (GE Life Sciences) equilibrated in HisTrap Buffer A. The column was washed with five column volumes of HisTrap Buffer A, and protein was eluted with HisTrap Buffer B (50 mM Tris-HCl, pH 7.5, 10 mM-mercaptoethanol, 250 mM imidazole). Fractions containing Pol I were then loaded onto a 5 *mL* HiTrap Heparin HP column (GE Life Sciences) equilibrated in Heparin Buffer A (50 mM Tris-HCl, pH 7.5, 1 mM DTT). Protein was eluted over a 0–50% gradient of Heparin Buffer B (50 mM Tris-HCl, pH 7.5, 1 mM DTT, 2 M NaCl). All purification steps were monitored by PAGE analysis (Appendix 2). Fractions containing Pol I were combined and exchanged into 50 mM sodium phosphate buffer, pH 7, using an Econo-Pac 10DG column (Bio-Rad) prior to labeling.

## Fluorophore labeling

Pol I or KF constructs were labeled with Alexa Fluor A594 or Alexa Fluor A488 C5 maleimide (ThermoFisher Scientific) and purified as described previously (*Berezhna et al., 2012*). Protein concentrations and labeling efficiency were calculated based on optical absorption using an extinction coefficient of $\varepsilon_{280}$ = 86,180 M$^{-1}$ cm$^{-1}$, $\varepsilon_{590}$ = 90,000 M$^{-1}$ cm$^{-1}$, and $\varepsilon_{495}$ = 71,000 M$^{-1}$ cm$^{-1}$ for Pol I, A594, and A488 respectively. Labeling efficiency was typically between 65 and 100%. Purified labelled protein was concentrated using a 50 kDa MWCO centrifugal filter (EMD Millipore) and stored at −80° C in buffer containing 10 mM Tris-HCl, pH 7.5, 1 mM EDTA, 1 mM DTT, and 50% (v/v) glycerol.

## smFRET data acquisition

smFRET data collection was performed using a custom built prism-based TIRF microscope as described previously (*Berezhna et al., 2012*). Briefly, a sample chamber was assembled with quartz slides passivated with polyethylene glycol and coated with neutravidin (*Lamichhane et al., 2010*). Biotinylated DNA substrates (20 pM) were flowed into the sample chamber and allowed to equilibrate for 5 min. The sample chamber was washed to remove unbound substrate, and 5 nM Pol I supplemented with 1 mM propyl gallate was introduced into the chamber. All solutions were prepared in imaging buffer (50 mM Tris-HCl, pH 7.5, 10 mM MgCl$_2$, 1 mM DTT, 50 mM NaCl, and 2 mM Trolox). Data were collected with 100 ms integration time using a custom single-molecule data

acquisition program that controlled the CCD camera. Single-molecule donor and corresponding acceptor time trajectories were extracted from movies using custom scripts written in IDL. The software packages to control the CCD camera and extract time trajectories were kindly provided by the laboratory of Dr. Taekjip Ha. All measurements were repeated at least six times by recording fluorescence data from different areas of the slide surface.

## smFRET data analysis

All data analyses were carried out using custom software written in-house using Python and supporting packages. Individual donor and corresponding acceptor intensity versus time traces were corrected for background signal by subtracting the median signal in each channel after photobleaching. Acceptor intensity traces were also corrected for leakage from the donor channel, as determined previously (*Berezhna et al., 2012*). FRET efficiency trajectories were calculated as

$$E = \frac{I_A}{I_A + I_D} \tag{1}$$

where $E$ is the apparent FRET efficiency at each time point and $I_D$ and $I_A$ are the corresponding corrected donor and acceptor intensities, respectively. Traces exhibiting anti-correlated fluctuations in the donor and acceptor intensities, constant total intensity (sum of donor and acceptor), and single-step photobleaching events were selected for further analysis.

Multiple FRET trajectories were analyzed globally using a Hidden Markov Model (HMM). The model was trained on all selected traces for a particular protein/substrate combination simultaneously using an expectation-maximization method (*Rabiner, 1989*). For each model, the minimum number of states that adequately fit individual traces was determined by manual inspection. Once the model was trained, the Viterbi algorithm (*Rabiner, 1989*) was used to determine the most likely hidden state path for each trajectory. This labelled state path was then used to aggregate all data points belonging to a particular state in order to compile composite histograms of FRET efficiency, using a Kernel Density estimation (KDE) algorithm (scikit-learn) with a Gaussian kernel and a bandwidth of 0.04. The relative populations of distinct FRET states were directly obtained during compilation of the corresponding histograms. Transition density plots (*McKinney et al., 2006*) were constructed using a KDE (Gaussian kernel, 0.04 bandwidth), where 2D points in the training data set were defined as the median FRET efficiency in the initial and final states. Dwell-time histograms were constructed with equal bin widths across the entire data range. The optimal bin width for each histogram was estimated using the Freedman-Diaconis rule

$$w = 2 \ \ \mathrm{IQR}(x) N^{-\frac{1}{3}} \tag{2}$$

where $w$ is the bin width, $x$ is the array of dwell times, $N$ is the number of data points in $x$ and $IQR$ is the interquartile range of the data.

Kinetic rate constants for overall decay of state $P$ ($k_{obs}^P$) were determined by fitting dwell-time histograms with a single-exponential function (SciPy). This observed rate constant is the sum of the microscopic rate constants for all available transitions ($k_{P \to N}$ and $k_{P \to U}$):

$$k_{obs}^P = k_{P \to N} + k_{P \to U} \tag{3}$$

where state U denotes unbound DNA.

The ratio of these rate constants is governed by the statistical frequencies of the corresponding transitions ($f_{P \to N}$ and $f_{P \to U}$):

$$\frac{k_{P \to N}}{k_{P \to U}} = \frac{f_{P \to N}}{f_{P \to U}} \tag{4}$$

The values of the transition frequencies and their associated uncertainties were estimated using a bootstrapping method. The number of relevant transitions were counted for a sample of 50 randomly selected traces from the total dataset. This process was repeated 50 times and the mean values of $f_{P \to N}$ and $f_{P \to U}$, and their associated standard deviations, were obtained. *Equations 3 and 4* were then solved to determine the microscopic rate constants, with errors propagated accordingly. The same analysis was applied to state $N$.

Overall $K_d$ values for binding of Pol I to DNA substrates were calculated according to

$$K_d = \frac{[\text{Pol}]_T}{f_B} - [\text{Pol}]_T \tag{5}$$

where $[\text{Pol}]_T$ is the total concentration of Pol I and the fraction of bound DNA, $f_B$, is defined as

$$f_B = \frac{\#\,\text{frames in any bound state}}{\text{total}\,\#\,\text{of frames}} \tag{6}$$

The mean value of $f_B$ and its uncertainty were determined using a similar bootstrapping method to that described above.

## Acknowledgements

We thank Edwin van der Schans for technical assistance with expression and purification of KF and Ashok Deniz for the use of the steady-state fluorimeter. This work was supported by the National Institutes of Health [R01 GM044060 to DPM, F32 GM115017 and T32 AI007354 to RFP].

## Additional information

### Funding

| Funder | Grant reference number | Author |
| --- | --- | --- |
| National Institute of General Medical Sciences | RO1 GM044060 | David P Millar |
| National Institute of General Medical Sciences | F32 GM115017 | Raymond F Pauszek III |
| National Institute of Allergy and Infectious Diseases | T32 AI007354 | Raymond F Pauszek III |

The funders had no role in study design, data collection and interpretation, or the decision to submit the work for publication.

### Author contributions

Raymond F Pauszek III, Software, Formal analysis, Investigation, Methodology, Writing - original draft, Writing - review and editing; Rajan Lamichhane, Formal analysis, Methodology; Arishma Rajkarnikar Singh, Resources, Methodology; David P Millar, Conceptualization, Formal analysis, Supervision, Funding acquisition, Methodology, Writing - original draft, Project administration, Writing - review and editing

### Author ORCIDs

Raymond F Pauszek III https://orcid.org/0000-0002-3445-8429
Rajan Lamichhane https://orcid.org/0000-0003-0089-6452
David P Millar https://orcid.org/0000-0001-9207-6958

### Decision letter and Author response

Decision letter https://doi.org/10.7554/eLife.62046.sa1
Author response https://doi.org/10.7554/eLife.62046.sa2

## Additional files

### Supplementary files

• Transparent reporting form

## Data availability

All graphical and tabular data are included in the manuscript and supplementary files. Single-molecule FRET trace data in HDF format have been deposited at zenodo (DOI https://doi.org/10.5281/zenodo.4555917).

The following dataset was generated:

| Author(s) | Year | Dataset title | Dataset URL | Database and Identifier |
|---|---|---|---|---|
| Pauszek RF, Lamichhane R, Rajkarnikar Singh A, Millar DP | 2021 | Single-molecule view of coordination in a multi-functional DNA polymerase | https://doi.org/10.5281/zenodo.4555917 | Zenodo, 10.5281/zenodo.4555917 |

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

# Appendix 1

The following tables present the various DNA oligos used to make each substrate in the current study. Each row lists (from top to bottom) the primer, template, and downstream strands for each substrate. B indicates the terminal biotin group, Y represents the amino-modified thymine used for fluorophore labeling, while lowercase nucleotides mark mismatched (flap) regions in the primer and downstream strands. All substrates were prepared by heat-annealing a mixture of oligos at 95° C for 5 min, followed by cooling slowly to room temperature on the lab bench. All mixtures contained a 3:1:3 molar ratio of primer, template, and downstream strands in order to ensure that all immobilized substrates contained all strands.

**Appendix 1—table 1.** Sequences of DNA oligonucleotides used to construct substrates for FRET scheme 1.

| Substrate | Oligo name | | Sequence |
|---|---|---|---|
| Double flap | 17e | 5'- | TCGCAGCCGYCAATATg |
| | T2b | 3'-BT$_{10}$- | AGCGTCGGCAGTTATAGATATAGCTTCGGAACAC −5' |
| | D18c_2 | | taCTATATCGAAGCCTTGTG −3' |
| Mixed flap | 17e | 5'- | TCGCAGCCGYCAATATG |
| | T1b | 3'-BT$_{10}$- | AGCGTCGGCAGTTATACATATAGCTTCGGAACAC −5' |
| | D18b_2 | | taGTATATCGAAGCCTTGTG −3' |
| Downstream flap | 17e | 5'- | TCGCAGCCGYCAATATG |
| | T1b | 3'-BT$_{10}$- | AGCGTCGGCAGTTATACATATAGCTTCGGAACAC −5' |
| | D18c_2 | | tacTATATCGAAGCCTTGTG −3' |
| Nick | 17e | 5'- | TCGCAGCCGYCAATATG |
| | T1b | 3'-BT$_{10}$- | AGCGTCGGCAGTTATACATATAGCTTCGGAACAC −5' |
| | D17b | | TATATCGAAGCCTTGTG −3' |
| 1nt gap | 17e | 5'- | TCGCAGCCGYCAATATG |
| | T1b | 3'-BT$_{10}$- | AGCGTCGGCAGTTATACATATAGCTTCGGAACAC −5' |
| | D16b | | ATATCGAAGCCTTGTG −3' |
| 2nt gap | 17e | 5'- | TCGCAGCCGYCAATATG |
| | T1b | 3'-BT$_{10}$- | AGCGTCGGCAGTTATACATATAGCTTCGGAACAC −5' |
| | D15b | | TATCGAAGCCTTGTG −3' |
| 3nt gap | 17e | 5'- | TCGCAGCCGYCAATATG |
| | T1b | 3'-BT$_{10}$- | AGCGTCGGCAGTTATACATATAGCTTCGGAACAC −5' |
| | D14b | | ATCGAAGCCTTGTG −3' |
| 4nt gap | 17e | 5'- | TCGCAGCCGYCAATATG |
| | T1b | 3'-BT$_{10}$- | AGCGTCGGCAGTTATACATATAGCTTCGGAACAC −5' |
| | D13b | | TCGAAGCCTTGTG −3' |
| Primer/template | 17e | 5'- | TCGCAGCCGYCAATATG |
| | T1b | 3'-BT$_{10}$- | AGCGTCGGCAGTTATACATATAGCTTCGGAACAC −5' |
| Primer/template-mismatch | 17e | 5'- | TCGCAGCCGYCAATATg |
| | T2b | 3'-BT$_{10}$- | AGCGTCGGCAGTTATAGATATAGCTTCGGAACAC −5' |

**Appendix 1—table 2.** Sequences of DNA oligonucleotides used to construct substrates for FRET scheme 2.

| Substrate | Oligo name | | Sequence |
|---|---|---|---|
| Double flap | P18mmt | 5'- | ACTTGAGAGCCGTATGAg |
| | T33Y | 3'-BT$_{10}$- | TGAACTCTCGGCATACTGGTAAYGGTAGCGTGC −5' |
| | D16 | | ttCCATTACCATCGCACG −3' |
| Mixed flap | P18 | 5'- | ACTTGAGAGCCGTATGAC |
| | T33Y | 3'-BT$_{10}$- | TGAACTCTCGGCATACTGGTAAYGGTAGCGTGC −5' |
| | D16 | | ttCCATTACCATCGCACG −3' |
| Downstream flap | P18 | 5'- | ACTTGAGAGCCGTATGAC |
| | T33Y | 3'-BT$_{10}$- | TGAACTCTCGGCATACTGGTAAYGGTAGCGTGC −5' |
| | D16mmt | | ttgCATTACCATCGCACG −3' |
| Primer/template | P1 | 5'- | ACTTGAGAGCCGTATG |
| | T33Y | 3'-BT$_{10}$- | TGAACTCTCGGCATACTGGTAAYGGTAGCGTGC −5' |

## Appendix 2

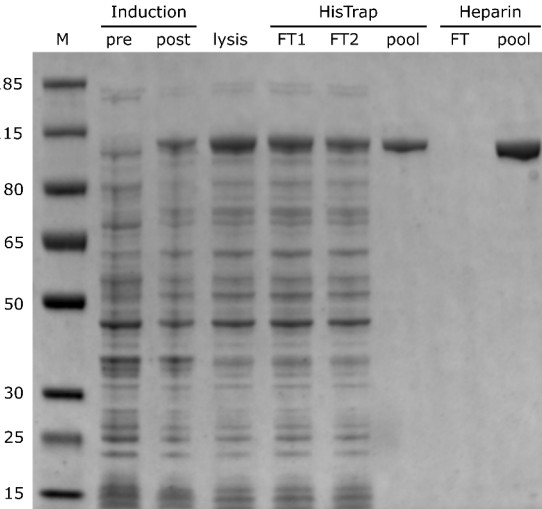

**Appendix 2—figure 1.** PAGE analysis of Pol I expression and purification steps. (M = size marker; FT = flow-through).

## Appendix 3

**Appendix 3—table 1.** Binding of Pol I to flap-containing DNA substrates.

| Substrate | $f_B$ | $K_d$ (nM) |
|---|---|---|
| Double flap | 0.28 ± 0.03 | 13 ± 2 |
| Mixed flap | 0.27 ± 0.04 | 14 ± 3 |
| Downstream flap | 0.22 ± 0.03 | 18 ± 3 |

$f_B$ is fraction of DNA bound by Pol I, determined as described in Materials and methods.

**Appendix 3—table 2.** Statistical frequencies of state-to-state transitions for Pol I interacting with flap-containing substrates.

| | Transition frequencies | | | |
|---|---|---|---|---|
| Substrate | $f_{P \to N}$ | $f_{P \to U}$ | $f_{N \to P}$ | $f_{N \to U}$ |
| Double flap | 0.6 ± 0.1 | 0.4 ± 0.1 | 0.4 ± 0.1 | 0.6 ± 0.1 |
| Mixed flap | 0.4 ± 0.1 | 0.6 ± 0.1 | 0.4 ± 0.1 | 0.6 ± 0.1 |
| Downstream flap | 0.34 ± 0.04 | 0.66 ± 0.04 | 0.4 ± 0.1 | 0.6 ± 0.1 |

**Appendix 3—table 3.** Rate constants for state-to-state transitions for Pol I interacting with flap-containing substrates.

| | Rate constants (s$^{-1}$) | | | |
|---|---|---|---|---|
| Substrate | $k_{P \to N}$ | $k_{P \to U}$ | $k_{N \to P}$ | $k_{N \to U}$ |
| Double flap | 1.2 ± 0.1 | 0.6 ± 0.2 | 0.56 ± 0.05 | 1.0 ± 0.2 |
| Mixed flap | 0.58 ± 0.04 | 0.8 ± 0.2 | 0.8 ± 0.1 | 1.3 ± 0.3 |
| Downstream flap | 0.34 ± 0.02 | 0.7 ± 0.1 | 1.6 ± 0.1 | 2.1 ± 0.4 |

**Appendix 3—table 4.** Statistical frequencies of state-to-state transitions for Pol I interacting with nick- or gap-containing substrates.

| | Transition frequencies | | | |
|---|---|---|---|---|
| Substrate | $f_{P \to N}$ | $f_{P \to U}$ | $f_{N \to P}$ | $f_{N \to U}$ |
| Nick | 0.4 ± 0.1 | 0.6 ± 0.1 | 0.5 ± 0.1 | 0.5 ± 0.1 |
| 1nt gap | 0.64 ± 0.04 | 0.36 ± 0.04 | 0.67 ± 0.04 | 0.33 ± 0.04 |
| 2nt gap | 0.6 ± 0.1 | 0.4 ± 0.1 | 0.7 ± 0.1 | 0.3 ± 0.1 |
| 3nt gap | 0.6 ± 0.1 | 0.4 ± 0.1 | 0.66 ± 0.04 | 0.34 ± 0.04 |
| 4nt gap | 0.54 ± 0.05 | 0.46 ± 0.05 | 0.7 ± 0.1 | 0.3 ± 0.1 |

**Appendix 3—table 5.** Rate constants for state-to-state transitions for Pol I interacting with nick- or gap-containing substrates.

| | Rate constants (/s) | | | |
|---|---|---|---|---|
| Substrate | $k_{P \to N}$ | $k_{P \to U}$ | $k_{N \to P}$ | $k_{N \to U}$ |
| Nick | 0.47 ± 0.02 | 0.6 ± 0.1 | 1.5 ± 0.1 | 1.7 ± 0.3 |
| 1nt gap | 0.52 ± 0.02 | 0.30 ± 0.04 | 1.6 ± 0.1 | 0.8 ± 0.1 |
| 2nt gap | 0.34 ± 0.02 | 0.23 ± 0.04 | 1.3 ± 0.1 | 0.6 ± 0.1 |

*Continued on next page*

*Appendix 3—table 5 continued*

| | Rate constants (/s) | | | |
|---|---|---|---|---|
| 3nt gap | 0.32 ± 0.02 | 0.23 ± 0.04 | 1.14 ± 0.05 | 0.6 ± 0.1 |
| 4nt gap | 0.20 ± 0.02 | 0.17 ± 0.03 | 1.4 ± 0.1 | 0.6 ± 0.2 |

**Appendix 3—table 6.** Binding of Pol I to various DNA substrates.

| Substrate | $f_B$ | $K_d$ (nM) |
|---|---|---|
| Nick | 0.20 ± 0.02 | 20 ± 3 |
| 1nt gap | 0.39 ± 0.02 | 8 ± 1 |
| 2nt gap | 0.47 ± 0.04 | 6 ± 1 |
| 3nt gap | 0.52 ± 0.05 | 5 ± 1 |
| 4nt gap | 0.56 ± 0.04 | 4 ± 1 |
| primer/template | 0.27 ± 0.02 | 14 ± 2 |

## Appendix 4

Control experiments were performed to determine whether there are any changes in spectroscopic parameters between states P and N that could alter the Förster radius and thereby cause changes in FRET efficiency. Steady-state emission spectra and polarization anisotropy values were acquired for A488-labeled DNA substrates, either alone or in the presence of saturating concentration of unlabeled Pol I. Similarly, emission spectra and anisotropy values were acquired for A594-labeled Pol I (labeled at position 550), either free in solution or in the presence of excess unlabeled DNA. The results are summarized in *Appendix 4—table 1*.

**Appendix 4—table 1.** Steady-state fluorescence controls.

| Labeled macromolecule | Label | Ligand | Normalized intensity [a,b] | Anisotropy [a] |
|---|---|---|---|---|
| Double flap | A488 | none | 1.0 | 0.05 ± 0.01 |
| | | Pol I | 0.94 | 0.21 ± 0.05 |
| Primer/template | A488 | none | 1.0 | 0.05 ± 0.01 |
| | | Pol I | 1.2 | 0.22 ± 0.01 |
| Pol I | A594 | none | 1.0 | 0.32 ± 0.02 |
| | | double flap | 1.2 | 0.34 ± 0.03 |
| | | primer/template | 1.1 | 0.34 ± 0.03 |

[a] total emission intensity normalized to the value for the free macromolecule [b] A488 data obtained with excitation at 495 nm, A594 with excitation at 590 nm

The primer/template exhibits a small increase in total emission intensity of A488 in the presence of unlabeled Pol I. In contrast, there is essentially no change in A488 intensity upon binding of Pol I to the double flap DNA substrate. The A488 anisotropy increases significantly in the presence of Pol I, as expected for binding of a large protein to DNA, but the final anisotropy values are similar for each bound substrate. Hence, the local rotational mobility of A488 must be similar for each bound DNA.

The emission intensity of A594 attached to Pol I shows little change upon binding of any of the DNA substrates, indicating that the local environment of A594 is unchanged. Likewise, the anisotropy of A594 is very similar in all bound complexes, indicating that the local rotational mobility of the probe is also unchanged.

The Förster radius ($R_0$) is dependent on the donor quantum yield ($\varphi_D$), the spectral overlap of donor and acceptor ($J$), the orientation factor ($\kappa^2$), and other parameters, as described by the following equation

$$R_0 = 9780\left(n^{-4}\kappa^2\varphi_D J\right)^{\frac{1}{6}} \tag{7}$$

where $n$ is the refractive index of the surrounding medium and $R_0$ is in units of angstroms (*Lakowicz, 2006*). Since the donor and acceptor anisotropies are very similar in each DNA-protein complex, it is likely that the orientation factor has the same value in each case. The spectral overlap must also be similar for each complex, since we do not observe any spectral shifts in donor emission or acceptor absorbance (not shown). The one quantity that does vary to some degree among the complexes is the donor quantum yield. Assuming an intrinsic $R_0$ value of 55.6 Å for the A488/A594 pair (*Gansen et al., 2018*), and using the normalized donor intensities in *Appendix 4—table 1*, we predict $R_0$ values of 57.0 Å for the bound primer/template and 55.0 Å for the double flap substrate. These reflect the $R_0$ values for states P and N, respectively, since the substrates mostly populate either state P (*Figure 4B*) or state N (*Figure 3C*). Overall, we conclude that the Förster radii for states P and N are very similar.

The apparent donor/acceptor distance $R$ corresponding to FRET efficiency $E$ was calculated as follows

$$R = R_0 \left( \frac{1}{E} - 1 \right)^{\frac{1}{6}} \tag{8}$$

