## [Decision Letter]

**Acceptance summary:**

Pauszek and colleagues present a number of well executed single-molecule experiments that use *E. coli* Pol I as a model DNA polymerase to demonstrate domain swapping for its various activities (i.e. DNA synthesis and 5' nuclease). As DNA pol I consists of two main units, one featuring domains for 3' to 5' DNA polymerisation and for 3' to 5' exonuclease activity, and the second containing 5' to 3' nuclease (*5' nuc* domain), understanding the coordination between these two parts and their three enzymatic activities is of great importance. The most significant results of this study include (1) the observation of reversible swapping between the DNA synthesis domain and 5' nuclease domain during a single encounter with DNA substrate and (2) there are varying positions of the 5' nuclease domain depending on the nature of the DNA substrate (i.e. gap, 5' flap, double flap). Previously, the authors applied similar experimental strategy to Klenow fragment. Comparing the data for KF and the full-length Pol I shows that nuc domain appears to suppress or override *pol-exo* transitions observed in KF.

**Decision letter after peer review:**

Thank you for submitting your article "Single-molecule view of coordination in a multi-functional DNA polymerase" for consideration by *eLife*. Your article has been reviewed by three peer reviewers, and the evaluation has been overseen by Maria Spies as a Reviewing Editor and José Faraldo-Gómez as the Senior Editor. The following individual involved in review of your submission has agreed to reveal their identity: Harold Kim (Reviewer #3).

The reviewers have discussed the reviews with one another and the Reviewing Editor has drafted this decision to help you prepare a revised submission.

Summary:

Pauszek and colleagues present a number of well executed single-molecule experiments that use *E. coli* Pol I as a model DNA polymerase to demonstrate domain swapping for its various activities (i.e. DNA synthesis and 5' nuclease). As DNA pol I consists of two main units, one featuring domains for 3' to 5' DNA polymerisation and for 3' to 5' exonuclease activity, and the second containing 5' to 3' nuclease (*5' nuc* domain), understanding the coordination between these two parts and their three enzymatic activities is of great importance. The authors chose a comprehensive number of DNA substrates that feature various flaps (downstream, double, mixed), gaps, and constructs without downstream DNA. The most significant results of this study include (1) the observation of reversible swapping between the DNA synthesis domain and 5' nuclease domain during a single encounter with DNA substrate and (2) there are varying positions of the 5' nuclease domain depending on the nature of the DNA substrate (i.e. gap, 5' flap, double flap). Previously, the authors applied similar experimental strategy to Klenow fragment. Comparing the data for KF and the full-length Pol I shows that nuc domain appears to suppress or override *pol-exo* transitions observed in KF. Overall, the manuscript is well written and presented, and supported by excellent experimental data.

The reviewers agree that the study is interesting, important and well executed. They have, however, identified several points that need to be addressed prior to publication:

Essential revisions:

1) The current study is performed under equilibrium conditions meaning that all potential enzymatic activities were abolished using suitable mutations. Moreover, the measurements were performed in absence of any nucleotides, which, depending on their complementarity to the free base of the template strand (if existing), are likely to influence the binding equilibrium of DNA pol I. Please comment on this and also on whether the DNA binding abilities of Pol I mutant (C262S/C907S/K550C/D424A/D116A, C262S/C907S/ /D424A/D116A/T213C) Pol and *5' nuc* domains are similar to that of the WT polymerase.

2) Figure 3C: The L361A mutation compared to WT Pol I histograms seem to indicate that the 0.6 FRET population does not predominately arise from binding of DNA to the exo domain. However, there is a significant shift in the two populations of FRET states, with the L361A mutant having 15% more in the Pol domain bound, or 0.8 FRET efficiency state with the double flap substrate. Can the authors speculate why this may be the case, if not related to exo domain DNA binding? Has this mutant been tested for exo binding in Pol I or just KF?

3) The representation of the histogrammed FRET data is unusual. The analysis of time traces using HMM is clear. Per trace, the authors end up with three FRET values, e.g., 0, 0.61 and 0.78. Summing up over all traces, the authors can calculate three mean values, e.g., 0, 0.6, 0.8. How do the authors then end up with the smooth curves shown in Figure 3D? Convoluting the mean FRET values with a log-Gaussian? To access the quality of the general experimental data, it would be informative to show the raw FRET histograms of all combined time traces in addition to the "smoothed" histograms in Figure 3C, Figure 4B, Figure 5C.

4) Please add some more example traces to the supplement. Whereas the histograms and transition plots of the mixed flap suggest a 50/50 chance of starting in either P or N mode, the example trace suggests all five binding encounters to start in the P mode. Please comment on that observation?

5) Subsection “Movement of DNA between *pol* and *5’ nuc* domains of Pol I”, seventh paragraph: "Interestingly,…"

Transfer rates were obtained from the exponential decay of the dwell time histograms (e.g., Figure 3—figure supplement 1). The fact that these apparent transfer rates P→N (kP→N) and P→U (kP→U) are similar is not a coincidence. Theoretically, the two rates should be similar (barring statistical uncertainty). P state can decay via two parallel pathways to either N or U. Therefore, the decay rate of the P state reflects the sum of the actual transfer rates (kP→N+kP→U) regardless of which state P decays into. To obtain the actual transfer rates (kP→N and kP→U), one should compute the relative frequency of transitions (how many decay to N vs. U as reported in Appendix 3—table 3), which determines the branching ratio kP→N/kP→U. From the sum and the ratio, individual rates kP→N and kP→U can be extracted. The rate constants reported in Appendix 3—tables 1, 2, 4 and 5 should be corrected, accordingly. Likewise, the apparent kP→N and kP→U should be similar for the mixed flap case as well. But the fact that they differ (Figure 3E) may suggest that the P state (0.8 FRET) measured for the mixed flap case is not a pure macrostate (or relatively impure compared to the downstream or double flap case). For example, the mixed flap P state can transition to N and U through different microstates that are all degenerate in their FRET values. This scenario makes a lot of sense because the mixed flap substrate should be in a dynamic equilibrium between different strand displacement intermediates.

6) Figure 3B: There seem to be short-lived events at a low FRET (e.g., at ~3 second). Could these represent unstable binding events like binding of the nuc domain to the downstream strand? How the frequency of these events changes with different substrates?

7) Are the apparent transition kinetics measured with the second labeling scheme similar to those measured with the first labeling scheme? Or perhaps is the data too noisy to extract rate constants?

---

## [Author Response]

Essential revisions:1) The current study is performed under equilibrium conditions meaning that all potential enzymatic activities were abolished using suitable mutations. Moreover, the measurements were performed in absence of any nucleotides, which, depending on their complementarity to the free base of the template strand (if existing), are likely to influence the binding equilibrium of DNA pol I. Please comment on this and also on whether the DNA binding abilities of Pol I mutant (C262S/C907S/K550C/D424A/D116A, C262S/C907S/ /D424A/D116A/T213C) Pol and 5' nuc domains are similar to that of the WT polymerase.

In response to the first point, the following sentences have been added to the Discussion section:

“The present study has been performed under equilibrium conditions, whereby the enzymatic activities of Pol I are disabled by mutations and nucleotide substrates are absent. […] Future smFRET studies with catalytically active Pol I and in the presence of nucleotides should provide further insights into the temporal order of events and the associated enzyme and DNA conformational changes during strand displacement synthesis and 5' flap processing.”

In response to the second point, the following sentence has been added to the Results section:

“Overall K_d_ values for binding of Pol I to the flap-containing DNA substrates (Appendix 3—table 1) are similar to values reported for KF interacting with primer/template substrates Christian, Romano and Rueda, 2009, Kuchta, Benkovic and Benkovic, 1988, and Polesky et al., 1990, indicating that the mutations within our Pol I construct do not disrupt the DNA binding ability of the protein.”

We have also explained in the Materials and methods section how K_d_ values were determined.

2) Figure 3C: The L361A mutation compared to WT Pol I histograms seem to indicate that the 0.6 FRET population does not predominately arise from binding of DNA to the exo domain. However, there is a significant shift in the two populations of FRET states, with the L361A mutant having 15% more in the Pol domain bound, or 0.8 FRET efficiency state with the double flap substrate. Can the authors speculate why this may be the case, if not related to exo domain DNA binding? Has this mutant been tested for exo binding in Pol I or just KF?

The 15% decrease in the 0.6 FRET population observed for the Pol I L361A mutant interacting with the double-flap substrate suggests that a small portion of the 0.6 FRET state arises from DNA binding to the *exo* domain in this case. Accordingly, we have added the following sentences to the Results section:

“However, for the double-flap substrate, the presence of the L361A mutation decreased the population of the 0.6 FRET state by 15% (Figure 3C), suggesting that this state arises, in part, from binding of the DNA to the *exo* domain. This is not unexpected, because the primer terminal base is mispaired with the template base in this DNA (Appendix 1—table1).”

In regard to the second point, we have indeed tested Pol I for *exo* site binding. We used a primer/template substrate containing the same terminal mismatch as in the double-flap substrate. The results are shown in a new figure (Figure 3—figure supplement 2). We have also added the following sentences to the Results section:

“Consistent with this, a primer/template substrate containing the same terminal mismatch (Appendix 1—table 1) exhibited a state at 0.6 FRET efficiency when bound to Pol I, and this state was completely eliminated with a L361A mutation within Pol I (Figure 3—figure supplement 2), confirming that mismatched DNA can engage the *exo* domain within Pol I.”

3) The representation of the histogrammed FRET data is unusual. The analysis of time traces using HMM is clear. Per trace, the authors end up with three FRET values, e.g., 0, 0.61 and 0.78. Summing up over all traces, the authors can calculate three mean values, e.g., 0, 0.6, 0.8. How do the authors then end up with the smooth curves shown in Figure 3D? Convoluting the mean FRET values with a log-Gaussian? To access the quality of the general experimental data, it would be informative to show the raw FRET histograms of all combined time traces in addition to the "smoothed" histograms in Figure 3C, Figure 4B, Figure 5C.

The smooth histogram curves in Figure 3C were obtained using a Kernel Density Estimation (KDE) algorithm with a Gaussian kernel and a bandwidth of 0.04, as described in the Materials and methods section. The histograms were *not* smoothed by convoluting mean FRET values with a log-Gaussian function. To illustrate the analysis, in Author response image 1, on the left we compare a conventional binned FRET efficiency histogram (blue bars) with the histogram obtained using the KDE algorithm (solid black line). Note that the exact same data (mixed flap DNA) was used in each case. It is clear that no information is lost during the KDE analysis. On the right, we show the separated histograms of states N (blue) and P (grey) from global KDE analysis, as presented in Figure 3C in the text.

4) Please add some more example traces to the supplement. Whereas the histograms and transition plots of the mixed flap suggest a 50/50 chance of starting in either P or N mode, the example trace suggests all five binding encounters to start in the P mode. Please comment on that observation?

Additional smFRET traces have been added in the new Figure 3—figure supplement 1. These show examples where Pol I initially encounters DNA in the N mode and later switches to the P mode, as well as other types of behavior.

5) Subsection “Movement of DNA between pol and 5’ nuc domains of Pol I”, seventh paragraph: "Interestingly,…"Transfer rates were obtained from the exponential decay of the dwell time histograms (e.g., Figure 3—figure supplement 1). The fact that these apparent transfer rates P→N (kP→N) and P→U (kP→U) are similar is not a coincidence. Theoretically, the two rates should be similar (barring statistical uncertainty). P state can decay via two parallel pathways to either N or U. Therefore, the decay rate of the P state reflects the sum of the actual transfer rates (kP→N+kP→U) regardless of which state P decays into. To obtain the actual transfer rates (kP→N and kP→U), one should compute the relative frequency of transitions (how many decay to N vs. U as reported in Appendix 3—table 3), which determines the branching ratio kP→N/kP→U. From the sum and the ratio, individual rates kP→N and kP→U can be extracted. The rate constants reported in Appendix 3—tables 1, 2, 4 and 5 should be corrected, accordingly. Likewise, the apparent kP→N and kP→U should be similar for the mixed flap case as well. But the fact that they differ (Figure 3E) may suggest that the P state (0.8 FRET) measured for the mixed flap case is not a pure macrostate (or relatively impure compared to the downstream or double flap case). For example, the mixed flap P state can transition to N and U through different microstates that are all degenerate in their FRET values. This scenario makes a lot of sense because the mixed flap substrate should be in a dynamic equilibrium between different strand displacement intermediates.

Following the reviewer’s suggestion, we have recalculated all the microscopic rate constants for the various state-to-state transitions. The revised values are presented in Figures 3E and 4D, and in Appendix 3—tables 3 and 5. We have also included the equations used to calculate the microscopic rate constants in the Materials and methods section.

6) Figure 3B: There seem to be short-lived events at a low FRET (e.g., at ~3 second). Could these represent unstable binding events like binding of the nuc domain to the downstream strand? How the frequency of these events changes with different substrates?

There does appear to be a transient binding event at ~3 s in Figure 3B, but this is not resolved and the final state (P or N) is unknown. Given this ambiguity, we have not attempted to count such events.

7) Are the apparent transition kinetics measured with the second labeling scheme similar to those measured with the first labeling scheme? Or perhaps is the data too noisy to extract rate constants?

In the second labeling scheme, the presence of 5 nM donor-labeled Pol I in solution necessarily increased the background level in the smTIRF recordings, resulting in relatively noisy traces. Accordingly, it was not possible to extract accurate rate constants for the various state-to-state transitions.